# COVID-19 Vaccine Acceptability and Adherence to Preventive Measures in Somalia: Results of an Online Survey

**DOI:** 10.3390/vaccines9060543

**Published:** 2021-05-21

**Authors:** Mohammed A. M. Ahmed, Robert Colebunders, Abdi A. Gele, Abdiqani A. Farah, Shariff Osman, Ibraahim Abdullahi Guled, Aweis Ahmed Moalim Abdullahi, Ahmed Mohamud Hussein, Abdiaziz Mohamed Ali, Joseph Nelson Siewe Fodjo

**Affiliations:** 1Department of Paediatrics, Faculty of Medicine and Surgery, Mogadishu University, Mogadishu P.O. Box 004, Somalia; ahmed.m@mu.edu.so (M.A.M.A.); shariff.osman@mu.edu.so (S.O.); ibrahimguled@mu.edu.so (I.A.G.); aweis@mu.edu.so (A.A.M.A.); ahmedmh@mu.edu.so (A.M.H.); drabdiaziz@mu.edu.so (A.M.A.); 2Department of Paediatric Cardiology, Uganda Heart Institute, Kampala P.O. Box 7051, Uganda; 3Global Health Institute, University of Antwerp, 2610 Antwerp, Belgium; robert.colebunders@uantwerpen.be; 4Norwegian Institute of Public Health, 0213 Oslo, Norway; abdi.gele@fhi.no; 5Department of Economics, Puntland State University, Wadajir, Garowe P.O. Box 090, Somalia; abdiqani.farah@psu.edu.so; 6Mogadishu Somali-Turkey Training and Research Hospital, Mogadishu P.O. Box 004, Somalia; 7De Martino Hospital, Hamar Jajab, Mogadishu P.O. Box 004, Somalia; 8Brain Research Africa Initiative (BRAIN), Yaoundé P.O. Box 25625, Cameroon

**Keywords:** COVID-19, vaccination hesitancy, Somalia, adherence, preventive measures

## Abstract

Most countries are currently gravitating towards vaccination as mainstay strategy to quell COVID-19 transmission. Between December 2020 and January 2021, we conducted a follow-up online survey in Somalia to monitor adherence to COVID-19 preventive measures, and COVID-19 vaccine acceptability and reasons for vaccine hesitancy. Adherence was measured via a composite adherence score based on four measures (physical distancing, face mask use, hand hygiene, and mouth covering when coughing/sneezing). We analyzed 4543 responses (mean age: 23.5 ± 6.4 years, 62.4% males). The mean adherence score during this survey was lower than the score during a similar survey in April 2020. A total of 76.8% of respondents were willing to receive the COVID-19 vaccine. Flu-like symptoms were more frequently reported in the current survey compared to previous surveys. Multiple logistic regression showed that participants who experienced flu-like symptoms, those in the healthcare sector, and those with higher adherence scores had higher odds for vaccine acceptability while being a female reduced the willingness to be vaccinated. In conclusion, our data suggest that the decreasing adherence to COVID-19 preventive measures may have caused increased flu-like symptoms over time. COVID-19 vaccine acceptance in Somalia is relatively high but could be improved by addressing factors that contribute to vaccine hesitancy.

## 1. Introduction

The long-term success of the coronavirus disease (COVID-19) public health response would rely on flattening the curve and reducing the burden on health systems until an adequate proportion of the population acquires immunity against the virus (herd immunity). Achieving population immunity by natural means (whereby a significant proportion of the population needs to become infected) would result in an unprecedented burden on healthcare services and could lead to 30 million deaths worldwide [1]. One key strategy to stop the escalation of the COVID-19 pandemic is to develop effective vaccines. So far, more than 100 COVID-19 vaccine candidates have been investigated. By 9th April 2020, five vaccine candidates were in phase 1 clinical trials and by November 2020, 30 vaccine candidates were in phase 3 clinical trials [2]. As of January 2021, three vaccines of Western pharmaceutical companies had been approved by international regulatory authorities and vaccination had started with the Chinese and Russian vaccines [3]. Currently COVID-19 vaccines are being administered in high-income as well as some middle-income countries. However, to stop the COVID-19 epidemic it is also important that large scale COVID-19 vaccination campaigns be initiated in low-income countries, including in sub-Saharan Africa (SSA). Given the large global population, and the observed high vaccine hesitancy coupled with relatively low vaccination coverage for currently available vaccines, understanding vaccine acceptance at the local level is critical [4]. Exploring predictors of vaccine attitudes in general terms has the potential to help policy makers recognize and adapt measures that improve vaccine trust that have previously been tested outside the COVID-19 pandemic [1].

Vaccines are widely recognized as a major tool for achieving public health success against the COVID-19 pandemic. Notwithstanding, certain groups of people may have doubts regarding the benefits of the available COVID-19 vaccines and concerns over their safety, causing them to question the need for vaccination; an attitude referred to as ‘vaccine hesitancy.’ Current literature shows disparities in vaccine acceptance across different geographical settings and population strata [5]. Some of the documented factors associated with COVID-19 vaccine acceptance in the general population include: vaccine effectiveness, adverse health effects, misinformation about the need for vaccination, lack of faith in the health system, lack of community awareness about vaccine-preventable diseases [5]. In China, higher vaccine acceptance rates were noted among healthcare workers (HCWs) compared to the general population [6]. Another study in the United States indicated that only one third of HCWs expressed willingness to receive the COVID-19 vaccine [7]. Globally, acceptance of COVID-19 vaccines was estimated to range from as low as 54.8% in Russia to as high as 88.6% in China; most Western countries showed relatively high public acceptance of the vaccine (59–75%) [8].

Besides vaccines, governments worldwide have been relying on large scale implementation of non-pharmaceutical interventions (such as physical distancing, quarantining, face mask use, hands and coughing hygiene) to curb COVID-19 transmission. Data from SSA reveal only moderate adherence to the public health measures instituted to prevent COVID-19, suggesting that a vaccine may be needed to overcome the pandemic in such settings. A study among HCWs in the Democratic Republic of Congo found that only 28% of them were willing to receive a COVID-19 vaccine [9]. Such a low acceptance could be blamed on an infodemic of misinformation and rumors that make it difficult to find credible sources of information [9]. In Somalia, where adherence to measures for COVID-19 prevention was seen to decrease over time during two consecutive online surveys [10], no data are available yet regarding the acceptance of a COVID-19 vaccine. Therefore, in a third round of online surveying, we sought to investigate both adherence to COVID-19 preventive measures and acceptability of a COVID-19 vaccine among Somalis, as artificial immunization may be the way forward in the face of decreasing observance of other recommended preventive practices.

## 2. Methods

### 2.1. Study Setting and Design

Between 26th December 2020 and 28th January 2021, we conducted a third cross-sectional online survey in Somalia to investigate observance of preventive measures and associated factors in Somalia. This study came as a follow-up of previous online surveys about adherence to COVID-19 preventive measures in Somalia, conducted in April 2020 and July 2020 [10]. Considering the study period (December 2020–January 2021) during which this third survey was conducted, 313 new confirmed COVID-19 cases and 19 deaths were officially reported [11]. Until January 2021, the number of new COVID-19 cases seemed to be on a consistent decline; however, by February 2021, a record number of new COVID-19 cases and related deaths were reported (Figure 1).

### 2.2. Survey Tool

The survey tool was an online questionnaire which was similar to that used during previous surveys by the International Citizen Project COVID-19 (ICPcovid) consortium [10]. The online questionnaire was adapted to the Somali context by the local investigators and translated it into English and Somali. All responses were submitted anonymously direct to the ICPcovid platform where they were stored in a secure website until data extraction. In addition to sociodemographic information and self-reported adherence to key COVID-19 preventive measures, this third survey included questions about vaccine acceptability. A copy of the full questionnaire used is available as Appendix A.

### 2.3. Recruitment of Study Participants

Respondents were recruited using a snowball sampling strategy, whereby the electronic survey link was widely disseminated via social media platforms (Twitter, WhatsApp, Facebook) and through the official website and media platforms of the Mogadishu University. Upon clicking on the link, users were introduced to a webpage containing information about the survey as well as the survey questions. Only data from consenting participants aged 18 years and older were analyzed. With a snowball sampling approach, we opted to reach as many respondents as possible during the study period.

### 2.4. Assessment of Adherence

We developed a composite adherence score, building on the respondents’ observance of the same COVID-19 preventive measures which were scored during the previous surveys in Somalia [10]: 1.5 m physical distancing, face mask use, hand hygiene (regularly washing hands with soap or using alcohol-based hand sanitizers), and coughing hygiene (covering one’s mouth when coughing). Adherence to a given measure was scored 1, and otherwise zero to obtain an overall score ranging from 0–4. In two previous surveys in Somalia, we included a fifth preventive measures “avoidance to touch one’s face.” Given that in this third survey we did not collect any data about touching one’s face, it was impossible to replicate the 5-item adherence score used during the first two surveys in Somalia.

### 2.5. Data Analysis

Continuous data were presented as mean and standard deviation (SD), while categorical data were summarized as percentages. The evolution of the degree of observance of COVID-19 preventive measures was assessed by comparing adherence scores across surveys and noting the number of new reported COVID-19 cases in Somalia [11] during the last week of each survey period. Group comparisons were performed using the Mann–Whitney U test and the Chi-Squared test as appropriate. Age- and sex-standardized values for vaccine acceptance were obtained by mapping the crude data unto the 2014 population structure of Somalia, as provided by the Somali National Bureau of Statistics [13]. A multiple logistic regression analysis was done to investigate factors associated with acceptance of the COVID-19 vaccine. All covariates yielding a *p*-value < 0.2 during univariate analysis were introduced in the final model. All tests were two-sided, and with a 5% significance threshold. Data analysis was done using the software R version 4.0.2.

### 2.6. Ethical Considerations

The study protocol was approved by the University of Antwerp Ethics Committee and the Mogadishu University’s Institutional Review Board. Informed e-consent (checkbox) was required from each participant before submitting responses. All responses were anonymous and securely stored in a password-protected server.

## 3. Results

Characteristics of Respondents

Responses from 4543 respondents were eligible for analysis during the third online survey. Participants resided in different regions of the country. The mean participant age was 23.5 ± 6.4 years, and 2837 (62.4%) participants were males (Table 1).

Overall mean adherence score (range: 0–4) during the third survey was comparable to that of the second survey (2.77 ± 1.26 vs. 2.76 ± 1.28; *p* = 0.879), but significantly lower than the initial survey in April 2020 (2.77 ± 1.26 vs. 2.85 ± 1.19; *p* = 0.020). Observance of different preventive measures varied significantly across surveys (Table 2). Of note, the age and sex structure of the study populations also varied significantly across surveys, though within a very narrow range (Table 2). In this third survey, mean adherence score was significantly lower among older age groups (>30 years), *p* < 0.001; mean adherence score among male and female respondents was similar (*p* = 0.113). Additionally, the mean adherence score among students in this third survey (representing 64.7% of our sample) was 2.82 ± 1.24, significantly higher than that of the other respondents (2.69 ± 1.29); *p* = 0.002. As for the internal consistency of the 4-item adherence score, we calculated Cronbach alpha (based on data from all three surveys) to be 0.669.

Regarding vaccine acceptance (Round 3 only; *n* = 4543), 3488 (76.8%) respondents expressed their willingness to receive a COVID-19 vaccine when it becomes available in Somalia. Reasons for vaccine refusal included: concerns about the vaccine being ineffective in 283 respondents, fear of side-effects in 424 respondents, confidence in a strong immune system that does not require a vaccine in 204 respondents, while another 308 respondents had the sentiment that the COVID-19 pandemic is over in Somalia hence no need for a vaccine. Other reasons for COVID-19 vaccine refusal recorded during this survey are documented in the Appendix A. Notably, the fear that some COVID-19 vaccines may contain substances derived from pigs was repeatedly mentioned as the reason for vaccine refusal. The age- and sex-standardized prevalence of vaccine acceptance in our survey was 73.0%. Vaccine willingness was not significantly different among students (76.5%) and other participants (77.2%); *p* = 0.631.

Compared to the previous survey findings in Somalia, flu-like symptoms were more frequently experienced during the past two weeks by respondents in the third survey (Table 3). Headache and fever were consistently reported as the most frequent symptoms across all surveys. During this third survey, 363 (8%) of respondents reported experiencing anosmia and/or ageusia; these symptoms were more frequent among those in the health sector (155/1636 [9.5%]) than among other participants (208/2907 [7.2%]); *p* = 0.007.

Figure 2 graphically summarizes the evolution of adherence across surveys (using mean adherence scores) and the COVID-19 burden in terms of weekly incidence at the end of each study period. We observed a drastic decrease in new COVID-19 cases between the first survey and the other two surveys.

The multiple logistic regression model found that the odds of accepting a COVID-19 vaccine were higher for respondents residing in Galmudug, Hirshabelle, or Southwest regions, those belonging to the healthcare sector, participants who had experienced flu-like symptoms in the past 14 days, and those with higher adherence scores. In contrast, being a female was associated with reduced odds of accepting the COVID-19 vaccine (Table 4).

## 4. Discussion

We conducted a follow-up online survey to assess the evolution of COVID-19 preventive behaviors in Somalia, as well as assess vaccine acceptability in the face of new COVID-19 waves around the globe. Although we acknowledge an important sampling bias in this survey (over 80% of respondents had university level education and resided in Benadir, and 64.5% were male), we report a rather encouraging level of vaccine acceptance of 76.8% (73.0% when age- and sex-standardized), higher than the 56% observed in the Democratic Republic of Congo using similar methods and with a similar study population as ours (>50% students) [15]. Regarding observance of preventive measures, the trend of decreasing adherence already noticed during the second survey [10] was maintained during this third survey. This observation could be in relation to the belief by many respondents that the COVID-19 outbreak is over in Somalia and normal lifestyle ought to resume. Another explanation could also be adherence fatigue by the population in view of the several months of altered lifestyle because of the pandemic. Paradoxically, a significant increase in flu-like symptoms was observed during the latter surveys suggesting that COVID-19 transmission is still rampant in Somalia. Symptoms such as anosmia and ageusia, which have an important diagnostic value when screening for suspected COVID-19 cases [14,16,17], were more frequent among participants from the healthcare sector. This confirms the fact that healthcare workers in Somalia stand a higher risk of contracting COVID-19, as already highlighted by empirical data [18]. There are suspicions that a new (possibly more virulent) strain of SARS-CoV-2 may be responsible for the February 2021 wave of COVID-19 in Somalia [19]. The fact that the current survey has identified fewer COVID-19 positive cases compared to the April 2020 survey could be as a result of under-testing and/or under-reporting [16].

About one quarter of respondents may refuse to be vaccinated for COVID-19 when the vaccine eventually becomes available in Somalia. Although the proportion of vaccine-compliant respondents is well above the targeted 67% required to achieve herd immunity (assuming a basic reproductive number R_0_ = 3 for COVID-19 transmission in Somalia) [15], the vaccine-hesitant individuals still represent a concern worth looking into, as they may eventually create clusters in which COVID-19 outbreaks can still occur, and they could even convince others to boycott vaccination campaigns. Although COVID-19 vaccination greatly reduces viral transmission, it is likely that even vaccinated persons may still transmit the coronavirus albeit to a lesser extent [20]; therefore it would be important to achieve the maximal attainable vaccine coverage. The only factor which was significantly associated with reduced odds of accepting the COVID-19 vaccination was the female gender. Women seem particularly affected by COVID-19 misinformation [21] (such as the rumor that the COVID-19 vaccine can render one sterile, considering that procreation is an important cultural mark of womanhood in African settings [22]). It is crucial that such erroneous beliefs and perceptions be addressed via intensive sensitization of the general population. Being a healthcare worker was associated with increased odds for vaccine acceptability in Somalia, in contrast with findings from the DRC [15]. Under such circumstances, healthcare workers could be leveraged as a means of sensitization since the general public usually trusts them and abide to their advice. Ultimately, more qualitative studies are required in the African setting to decipher and address possible causes of COVID-19 vaccine hesitancy [23]. It is worth noting that in Somalia as in other Muslim nations, the ‘halal’ nature of any vaccine is of prime importance [24]. This explains why concerns were raised regarding the use of porcine derivatives when manufacturing COVID-19 vaccines. Although information supporting this theory was widely disseminated via social media, to the best of our knowledge none of the COVID-19 vaccines approved till date contain any substance derived from pigs. All the same, it is crucial to take into account such cultural aspects when deciding which vaccines should be introduced in the country.

This study is not void of limitations. As mentioned earlier, our web-based approach has introduced non-negligible sampling bias such that the study population is far from being representative of the general population in Somalia. We acknowledge the sampling limitation which caused our study population to be biased towards students, whose adherence scores were significantly different from other respondents. Additionally, the fact that majority of participants resided in Benadir makes it difficult to generalize our observations to the entire country. However, COVID-19 vaccine acceptance rates among students were similar to acceptance rates of other survey participants, suggesting that similar acceptance rates could be expected in a less biased Somalian study population. Nevertheless, this survey should only be considered to be a starting point for further research concerning vaccine acceptability and reasons for vaccine hesitancy. Ideally, such studies should be done in a random sample of the population and qualitative research among different strata of the Somali society should be considered.

In conclusion, our data suggest that while adherence to COVID-19 preventive measures is on a gradual decrease, viral transmission seems to be on the rise as can be seen by an increase in flu-like symptoms over time. COVID-19 vaccine acceptance in Somalia is relatively high (compared to other African countries) and could be improved by addressing the factors that contribute to vaccine hesitancy, mainly via sensitization of the general public to dispel any misconceptions.

## Figures and Tables

**Figure 1 vaccines-09-00543-f001:**
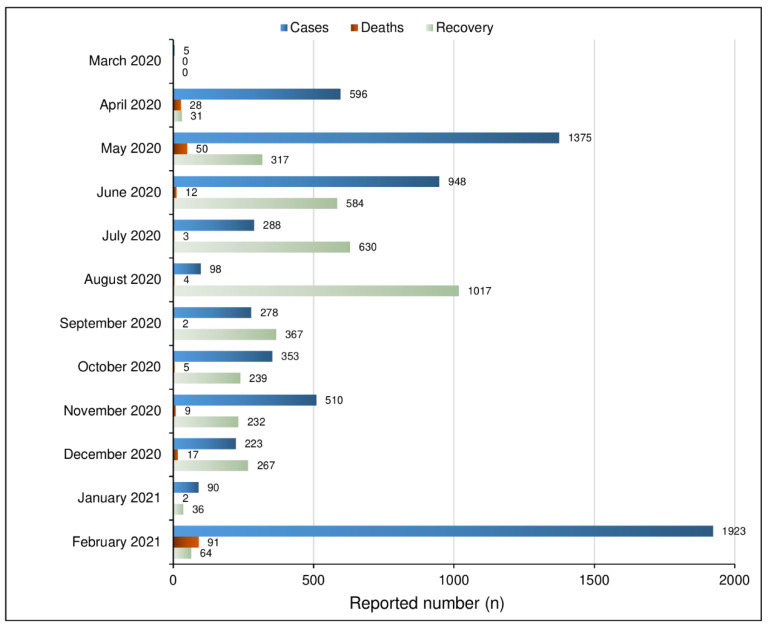
COVID-19 in Somalia Report Chart from March 2020 up to February 2021 [12].

**Figure 2 vaccines-09-00543-f002:**
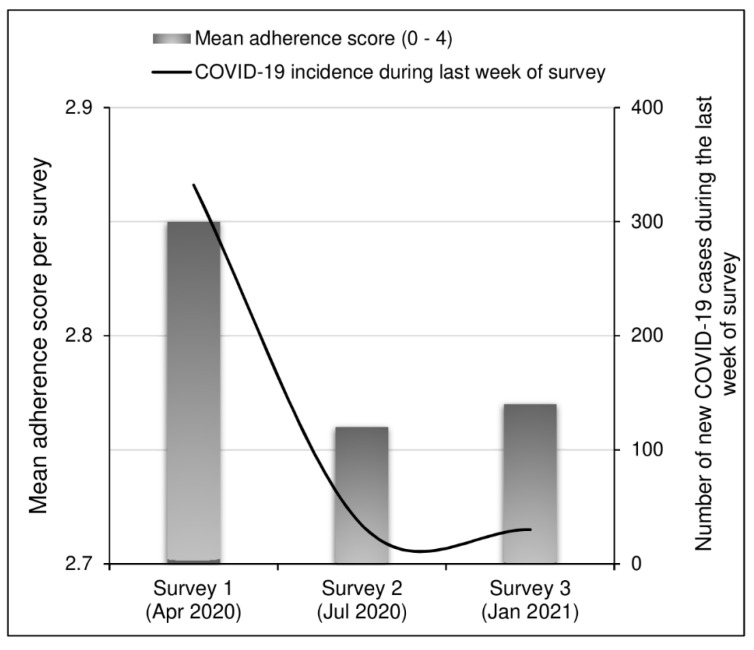
Mean adherence scores and COVID-19 incidence during the last week of each survey period.

**Table 1 vaccines-09-00543-t001:** Participants’ characteristics.

Characteristics	Survey Findings (*n* = 4543)
Age in years: Mean (SD)	23.5 (6.4)
Gender: *n* (%)	
Male	2837 (62.4%)
Female	1706 (37.6%)
Highest educational level: *n* (%)	
Primary	103 (2.3%)
Secondary	522 (11.5%)
University Undergraduate	3423 (75.3%)
University Postgraduate	495 (10.9%)
Region: *n* (%)	
Benadir	3721 (81.9%)
Galmudug	168 (3.7%)
Hirshabelle	262 (5.8%)
Jubaland	107 (2.4%)
Puntland	117 (2.6%)
Somaliland	67 (1.5%)
Southwest	101 (2.2%)
Residential setting: *n* (%)	
Rural	189 (4.2%)
Sub-urban	293 (6.5%)
Urban	4061 (89.4%)
Live alone in residence: *n* (%)	336 (7.4%)
Profession: n (%)	
Student	2941 (64.7%)
Jobless	554 (12.2%)
Self-employed	378 (8.3%)
Private company employee	540 (11.9%)
Government employee	130 (2.9%)
Student/worker in the health sector: *n* (%)	1636 (36.0%)
Working from home at the time of responding to this survey: *n* (%)	1289 (40.6%)
Suffer from a chronic/underlying condition: *n* (%) *	356 (7.8%)
Experienced flu-like symptoms during the past 14 days: *n* (%) **	2042 (44.9%)
Tested for COVID-19: *n* (%)	741 (16.3%)
Timing of COVID-19 test: n (%)	
During the past 14 days	230 (31.0%)
Between two weeks and one month ago	135 (18.2%)
More than one month ago	376 (50.7%)
Test results for those tested during the past 14 days: n (%)	
Positive	60 (27.9%)
Negative	155 (72.1%)
Not known	15

* Chronic conditions including heart disease, diabetes, hypertension, cancer, HIV, or asthma. ** Flu-like symptoms that fit in the WHO clinical definition for suspected COVID-19 [14].

**Table 2 vaccines-09-00543-t002:** Comparative characteristics of the study population and observance of COVID-19 preventive measures across surveys.

	Survey 1	Survey 2	Survey 3	*p*-Value
	*n* = 4124	*n* = 4703	*n* = 4543	
	(April 2020)	(July 2020)	(Jan 2021)	
**Sociodemographic characteristics: *n* (%)**				
Age group				<0.001
18–20 years	1455 (35.3%)	1506 (32.0%)	1617 (35.6%)	
21–30 years	2456 (59.6%)	2786 (59.2%)	2551 (56.2%)	
31–40 years	158 (3.8%)	297 (6.3%)	262 (5.8%)	
41–50 years	20 (0.5%)	65 (1.4%)	64 (1.4%)	
51–60 years	28 (0.7%)	32 (0.67%)	39 (0.9%)	
61 years and above	7 (0.2%)	17 (0.4%)	10 (0.2%)	
Gender				0.004
Male	2490 (60.5%)	2768 (59.1%)	2837 (62.4%)	
Female	1626 (39.5%)	1916 (40.9%)	1706 (37.6%)	
**COVID-19 preventive measures: *n*** **(%)**				
Mask use	2112 (51.2%)	2644 (56.2%)	2371 (52.2%)	<0.001
Observe 1.5 m physical distancing	2634 (63.9%)	2781 (59.1%)	2602 (57.3%)	<0.001
Wash hands regularly with soap	3334 (80.8%)	3491 (74.2%)	3560 (78.4%)	<0.001
Use alcohol-based hand gel regularly	2352 (57.0%)	2687 (57.1%)	2850 (62.7%)	<0.001
Cover mouth when coughing/sneezing	3606 (87.4%)	3875 (82.4%)	3937 (86.7%)	<0.001
Stay at home when feeling flu-like symptoms	3464 (84.0%)	3418 (72.7%)	3353 (73.8%)	<0.001
Travelled during the past 7 days	217 (5.3%)	827 (17.6%)	640 (14.1%)	<0.001
Adherence score category:				<0.001
0	193 (4.7%)	286 (6.1%)	279 (6.1%)	
1	454 (11.0%)	639 (13.6%)	545 (12.0%)	
2	728 (17.7%)	887 (18.9%)	915 (20.1%)	
3	1134 (27.5%)	990 (21.1%)	998 (22.0%)	
4	1615 (39.2%)	1901 (40.4%)	1806 (39.8%)	
Willing to receive COVID-19 vaccine			3488 (76.8%)	NA
Reasons for COVID-19 vaccine refusal *				NA
(N = 1055 respondents who were unwilling)				
Vaccine is ineffective			283 (26.8%)	
Fear of side-effects			424 (40.2%)	
COVID-19 is over, no need for vaccine			308 (29.2%)	
My immune system is already very strong			204 (19.3%)	
Other reasons **			29 (2.7%)	

* Many responses possible for each participant. ** See details in Appendix A.

**Table 3 vaccines-09-00543-t003:** Self-reported flu-like symptoms within the two weeks preceding each survey.

Symptoms: n (%) *	Survey 1	Survey 2	Survey 3	*p*-Value
	*n* = 4124	*n* = 4703	*n* = 4543	
Fever	360 (8.7%)	658 (14.0%)	1135 (25.0%)	<0.001
Headaches	442 (10.7%)	918 (19.5%)	1146 (25.2%)	<0.001
Dry cough	155 (3.8%)	230 (4.9%)	420 (9.2%)	<0.001
Productive cough	146 (3.5%)	194 (4.1%)	242 (5.3%)	<0.001
Sore throat	216 (5.2%)	356 (7.6%)	556 (12.2%)	<0.001
Coryza	300 (7.3%)	397 (8.4%)	531 (11.7%)	<0.001
Loss of smell (anosmia)	278 (6.7%)	345 (7.3%)	264 (5.8%)	0.012
Loss of taste (ageusia)	230 (5.6%)	282 (6.0%)	260 (5.7%)	0.690
Shortness of breath	86 (2.1%)	150 (3.2%)	139 (3.1%)	0.003
Myalgia	305 (7.4%)	345 (7.3%)	450 (9.9%)	<0.001
Fatigue	254 (6.2%)	251 (5.3%)	465 (10.2%)	<0.001
Nausea	119 (2.9%)	182 (3.9%)	207 (4.6%)	<0.001
Diarrhea	70 (1.7%)	126 (2.7%)	145 (3.2%)	<0.001

* Flu-like symptoms that fit in the WHO clinical definition for suspected COVID-19 [14].

**Table 4 vaccines-09-00543-t004:** Multiple logistic regression investigating determinants of COVID-19 vaccine acceptance (Round 3 data only).

Covariate	Adjusted OR (95% CI)	*p*-Value
Age	1.011 (0.997–1.025)	0.507
Gender		
Male	Reference	
Female	0.836 (0.718–0.973)	0.021
**Region**		
Benadir	Reference	
Galmudug	1.583 (1.036–2.504)	0.041
Hirshabelle	2.206 (1.498–3.361)	<0.001
Jubaland	1.084 (0.657–1.867)	0.761
Puntland	1.285 (0.826–2.061)	0.281
Somaliland	1.334 (0.734–2.579)	0.365
Southwest	2.038 (1.149–3.912)	0.022
**Residential setting**		
Rural	Reference	
Sub-Urban	1.272 (0.768–2.090)	0.345
Urban	1.168 (0.763–1.749)	0.461
**Profession**		
Student	Reference	
Unemployed	0.919 (0.736–1.152)	0.456
Self-employed	1.016 (0.764–1.363)	0.913
Private employee	0.978 (0.760–1.264)	0.861
Government employee	1.251 (0.753–2.170)	0.405
**Educational level**		
Primary	Reference	
Secondary	0.786 (0.420–1.407)	0.433
Undergraduate	0.670 (0.367–1.160)	0.169
Postgraduate	0.595 (0.315–1.075)	0.096
Student/worker in the healthcare sector	1.458 (1.235–1.725)	<0.001
Adherence score	1.540 (1.451–1.634)	<0.001
Presence of underlying disease	0.905 (0.677–1.223)	0.509
Presence of flu symptoms	1.521 (1.307–1.772)	<0.001
Has been tested for COVID-19	1.083 (0.867–1.360)	0.488

OR: Odds ratio; CI: Confidence interval.

## Data Availability

The data presented in this paper are available upon reasonable request to the corresponding author.

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
