# Peer review of "COVID-19 Vaccine Acceptability and Adherence to Preventive Measures in Somalia: Results of an Online Survey"

_vaccines, 2021, doi:10.3390/vaccines9060543_

Round 1

Reviewer 1 Report

You have accurately responded to the comments

Author Response

Thank you for your response.

Reviewer 2 Report

The authors have responded to the reviewer's suggestions.

However, because of the limited sample (all university students in one town) I suggest that it should be shortened.

I had previously suggested to the editor that the authors be invited to submit a shortened version, which can then be published.

The shortened version should be in the form of a research report, with the text words 50% of the current total 

Author Response

We propose not to shorten the paper as suggested by the reviewer. There is very limited information available about COVID-19 in Somalia. Therefore we believe it is important to publish all the scientific data we collected and this is possible in an online journal. These data could stimulate other researchers to plan additional more detailed and larger studies. Moreover, the 2 other reviewers did not propose to modify or shorten this revised version of the manuscript.

Reviewer 3 Report

The manuscript has been improved substantially. I just have a minor comment:

Figure 1: There seems to be no data for March 2020, so it can be removed from the figure. 
You can zoom into this figure to lengthen the bars.

Author Response

Fig 1 has now been revised.

This manuscript is a resubmission of an earlier submission. The following is a list of the peer review reports and author responses from that submission.

Round 1

Reviewer 1 Report

The manuscript presents the results of a survey regarding vaccine acceptance/refusal. The study is within the scope of the manuscript and adds news information. But, it needs several revisions.
Please see my comments below:

This is not clear. Please rephrase:
"130
An adherence score that included five preventive measures could not be used as in
131
the previous surveys because we did not assess whether the participant avoided touching
his/her face during this third round."

The last part of Table 1 has mistakes. Please correct it. Check the last column.

Reviewer 2 Report

The authors are to be commented for attempting to assess Sars vaccine acceptability in Somalia.

Methods:

The methods you used received responses from a sample consisting of undergraduates or postgraduates (86.2%), mostly males with 82% residing in one region (Benadir). 

Please state which symptoms you assessed correspond to the US CDC criteria.

Results: Because this is a highly limited and biased sample of 86% university students/graduates and mostly male the words "university students and postgraduates in Benadir region should appear in the title and abstract 

The article should be drastically shortened to a research note. 

Reviewer 3 Report

I was invited to revise the paper entitled "COVID-19 vaccine acceptability and adherence to preventive measures in Somalia: Results of an online survey". It aimed to investigate both adherence to COVID-19 preventive measures and acceptability of a COVID-19 vaccine among Somalis.

It was a well written article, that described a three round online survey among Somalis.

The study is interesting and improve the knowledge in this field.

I have only some minor observations:

  • Sample size estimation was not reported;
  • I suggest to test the adherence score also as continuous variables across three survey;
  • As supplementary material, please report the score variation by gender and age categories.

Reviewer 4 Report

This is an interesting paper using an online survey to investigate the adherence to preventive measurements to prevent spreading of COVID-19 and the willingness to accept a vaccination against SARS-COV2 virus infections in Somalia.

The fundamental problem with this online survey that there is a very large bias towards young people with higher education who have access to the technology and understand the significance of such surveys. The authors acknowledge this limitation, but still present the outcome as representative for the population of Somalia and compare their results with those in other countries and regions. The authors can correct to some extent for this bias and compare the outcomes of  the majority group of young academic professionals with the rest of respondents and see if there are significant differences between those groups. When they observe differences they can discuss about the reasons for those differences (e.g. educational levels or other differences). When they do not see significant differences this could be an argument that despite the bias different groups within Somalia share some of those opinions.

Secondly, a major disadvantage of this survey is that it is very much describing the situation at one particular moment in time and with the rapidly changing conditions with respect to infection pressure, availability of different vaccines, the results are already outdated at the moment that they are taken. The authors do compare the outcome of this survey with two previous taken surveys, but they do not give details about the population of the previous surveys so it unclear whether the same bias was true for the previous survey and possibly they  have to correct in a similar way for such a bias.